# Yoga as a Contemplative Practice and Its Contribution to Participatory Self-Knowledge and Student Retention: A Scoping Review of the First-Year Undergraduate Student Transition

**DOI:** 10.3390/ijerph21070884

**Published:** 2024-07-07

**Authors:** Beverley Martin, Blake Peck, Daniel Terry

**Affiliations:** 1Institute of Health and Wellbeing, Federation University Australia, 1 University Way, Mt. Helen, VIC 3350, Australia; beverley.martin@federation.edu.au (B.M.); b.peck@federation.edu.au (B.P.); 2School of Nursing and Midwifery, University of Southern Queensland, 11 Salisbury Rd, Ipswich, QLD 4305, Australia; 3Centre for Health Research, University of Southern Queensland, 11 Salisbury Rd, Ipswich, QLD 4305, Australia

**Keywords:** yoga, well-being, stress, anxiety, mindfulness, learning

## Abstract

Background: Contemplative pedagogy, specifically yoga, introduced into the higher education curriculum has the potential to develop and entrain intellectual, emotional, and social development in relation to mental health among university students studying for medical and nursing degrees. The objective of the study is to determine the extent of the current literature on the prevalence of yoga as a contemplative practice that contributes to student well-being and self-knowledge in the first-year transition from high school to university. Methods: As part of the scoping review, CINAHL, EBSCO, Medline, Emerald, Eric, and PsycINFO were searched to identify the prevalence and connection of mind–body courses to student well-being between 2011 and 2022. Screening and selection of studies were based on eligibility criteria and methodological quality assessment. Colaizzi’s method of data analysis enabled the phenomena of interest to be examined and follows the PRISMA for Scoping Reviews (PRISMA-ScR) checklist. Results: Seventeen studies were included with two themes emerging, which include physical practices and training and barriers to success. Conclusion: Yoga is a practice that supports undergraduate students in managing their stressful lives. Due to the experiential nature of yoga the participatory reflective processes established within the physicality of the students provided a framework to cope with the stress and challenges of higher education.

## 1. Introduction

Within the context of tertiary or higher education, university students have a propensity to experience increased levels of psychological distress, such as anxiety, depressive symptoms, and psychosocial issues of self-regulation, motivation, procrastination, and poor self-care [1,2,3]. Psychological, physiological, and psychosocial factors that university students experience have been demonstrated to contribute to academic difficulties and, if not acknowledged and addressed, may lead to poor academic success and student attrition [3,4].

Contemplative practices, such as yoga and mindfulness, have been conceptualized as forms of mental training intended to facilitate changes in cognitive and emotional processes and include self-regulation and attention training [5]. Specifically, the theoretical framework of this paper is centered on the premise that self-regulation through physical practice, such as yoga, is a potent strategy for preventing and managing psychosocial challenges. Within this framework, optimal self-regulation is conceptualized as the ability of an individual to sustain awareness and attend to the physiological, emotional, and cognitive needs of their inner self. Concurrently, they interact effectively with their familial, communal, and cultural contexts. This equilibrium between inner self-care and external engagement forms the crux of this theoretical framework. Within this context, contemplative practices, such as mindfulness and yoga, not only impact psychosocial challenges but also provide an education on the capacity to develop perspectives within an individual. Further, contemplative practices also provide a framework to examine assumptions and metacognitive abilities, which encompass the capacity to develop knowledge about oneself which is not directly observable [6,7].

Since the 1990s, there have been new opportunities for higher education students to enroll in contemplative education programs [8]. For example, Francisco Varela (1946–2001) led summer programs at Naropa University investigating cognitive science and the ‘first-person inquiry’ and wisdom born within individuals [9]. Further, the defining characteristics of contemplative practices, which include meditation and yoga, are that they operate directly on psychological functions such as attention and emotional regulation and may affect neural systems, psychological functions, and behavioral outcomes [10].

Extant evidence aligns with the benefits of contemplation for mental health and may aid students in terms of stress, sleep, depression, mindfulness, self-compassion, and quality of life [11]. Contemplative pedagogy encourages students to form a deeper and more personal relationship with what is being taught in the classroom, cultivating greater self-awareness and depth of knowledge [12]. Contemplative pedagogy, when introduced into the higher education curriculum, has the potential to develop and support the intellectual, emotional, and social development of university students, particularly in relation to mental health [10].

Within the literature, mindfulness research has increased exponentially over the past 20 years and is classified into three epochs. Within the first epoch (1916–1999), the focus was largely on theoretical and spiritual approaches. Epoch 2 (2000–2009) was a specific inquiry into mindfulness-based interventions, while the final epoch (2010–2019) was more involved in robust validation approaches [13].

Within this context, mindfulness practices have demonstrated they are often embedded with either the physical practice of yoga or mindfulness training as part of an overall intervention to support students [5]. As such, mindfulness is described as a quality of consciousness [14], which is an abiding state of present-moment awareness [15], and as an approach for increasing awareness and responding skillfully to mental processes that contribute to emotional distress and maladaptive behavior [16].

Previous reviews have highlighted the benefits of contemplative practices for university students. For example, Pascal and Marken [17], within their scoping review of qualitative mindfulness practice research, concluded mindfulness practices had so far not unleashed their full potential and the strong reliance on participants’ self-reports [17]. In contrast, Kinsella et al. [18] aimed to identify, summarize, and describe the current state of knowledge on mindfulness in allied health and social care professional education. Mindfulness education interventions included attention and concentration, empathy, and positive relationships and had the potential to make important contributions to the education of future practitioners [18]. Another study evaluated the preliminary feasibility, acceptability, and effects of an 8-week mindfulness curriculum for interprofessional healthcare professionals and trainees [19]. The course was purposefully designed to include both theory-driven and didactic content as well as hands-on experiences of mindfulness of mindfulness-based movement practices. There was significant interest in the course, with qualitative data supporting the feasibility of interprofessional mindfulness courses [19].

It is evident that engaging in contemplative practices develops and supports students in various ways during the first year of transition within higher education. However, this scoping review seeks to understand what impact yoga has on the psychological, psychosocial, and behavioral aspects of learning among first-year higher education students.

## 2. Purpose of the Study

Within the context of higher education, high levels of psychological distress, such as anxiety, depressive symptoms, and psychosocial issues of self-regulation, motivation, procrastination, and self-care are experienced by university students [1,2,3]. This has been shown to be further heightened during periods of confinement and virtual learning, such as during the global pandemic, but also as students are moving more and more to learning virtually and in isolation from their peers [20]. Psychological, physiological, and psychosocial factors may contribute to academic difficulties and, if not acknowledged and attended to, may result in students’ academic failure and attrition from university [4]. Within this context, the aim of this scoping review was to examine yoga and mindfulness practices that act as metacognitive practices and afford the cultivation of behavioral change, including self-regulation. Over time, opportunities exist for insight that restructure and reframe an individual’s sense of agency, primarily through reflective attention to experience and movement of the body.

## 3. Methods

### 3.1. Study Design

The scoping review method was planned and conducted in adherence to the Preferred Reporting Items for Systematic reviews and Meta-Analysis extension for Scoping Reviews (PRISMAScR) statement [21]. The structure of the approach was based on the theoretical framework for scoping reviews developed by Arksey and O’Malley [22]. Specifically, the five-step framework approach involves searching for relevant studies; the study selection; charting the data; and collating, summarizing, and reporting the results.

### 3.2. Search Strategy and Study Selection Process

The comprehensive review of the databases encompasses a preliminary consultation with an academic librarian specialist to ensure key database selection and accuracy and enable optimal outcomes of the review. The literature search was conducted between February and June 2022, with a follow-up search conducted in 2024 to ensure the recent literature was gleaned. As guided by the librarian specialist, EBSCO, CINAHL, Medline, Emerald, Eric, and PsycINFO databases were identified and searched for the peer-reviewed literature published between 2011 and 2022. In addition, the search engines Google and Google Scholar were also searched to identify any additional research that may have not been captured through the database searcher. Lastly, reference lists of relevant studies were also examined to identify any relevant studies not captured through the database and search engine searches.

The initial literature search enabled the author to identify the fundamental index terms and keywords from the main searches. To ensure inclusivity, keywords included “yoga” AND “learner” OR “student” AND “university” OR “college” OR “higher education” and were the main terms used in the database searches. Secondary search terms were combined with the Boolean operator OR and then in combination with additional terms such as “self-compassion” OR “yoga” AND “concentration” OR “attention” AND “study” to facilitate the recovery of relevant studies. All relevant keywords and their synonyms were used to develop search strings to increase search sensitivity and reduce the risk of relevant key studies being omitted. The search string focused on keywords in titles and abstracts and was used in the databases. Based on the eligibility criteria, titles and abstracts were screened. Agreement was reached between all authors regarding the search strings.

### 3.3. Inclusion and Exclusion Criteria

The search strategy was developed based on specific inclusion criteria, which included first-year university students. Articles were considered eligible for inclusion in this scoping review if they encompassed a mind–body practice that included yoga postures and mindfulness or if the intervention was restricted to first-year university students and included an evaluation of the skills related to mind–body practices suitable for first-year university students.

Articles were excluded if they were not original research; if they were non-intervention studies; if they were focused only on yogic breathing, meditation, and mindfulness practices with no physical body practice; if they emphasized body image or blood pressure only; if participants were post-graduate students; or if articles were not written in English.

### 3.4. Study Screening

The articles retrieved from the search were exported to EndNote (version X7). Titles and abstracts were initially screened independently by the first author and cross-checked by the second and third authors. To increase rigor and reliability, a second round of full-text articles was reviewed independently by the second and third authors. After screening and selecting titles and abstracts, eligible records were obtained as full texts. The screening and selection of the full-text articles were performed by the first, second, and third authors. Any disagreements about the inclusion or exclusion of studies that arose were resolved through discussion with the three authors. Once full agreement was achieved between the research team, full-text assessment was undertaken. The final list of included studies was evaluated and verified by the research team.

### 3.5. Methodological Assessment

Given the complexities and diversity of the literature that was gleaned, various methodological assessments were undertaken to ensure quality. This appraisal was undertaken among all identified publications to assess risk of bias, guided by checklists produced by the Critical Appraisal Skills Programme [23], the Best Evidence Medical Education (BEME) quality indicators [24,25], and the Joanna Briggs Institute (JBI) critical appraisal tool for randomized control trials [26].

The methodological quality assessment of the quantitative randomized controlled trials was evaluated according to the JBI critical appraisal tool for randomized control trials in systematic reviews [26]. The remaining quantitative papers underwent a methodological evaluation using the Best Evidence Medical Education (BEME) systematic review guide [25]. Studies of superior quality were those that satisfied at least seven out of the 11 indicators. In this context, all discovered articles were scrutinized using the BEME checklist, with each criterion classified as either “met” (+), “not met” (−), or “not applicable” (n/a). Each criterion of the BEME is then scored to provide an overall quality score ranging between 0 and 11, while those excluded had a score equal to or less than 6 [25] (Appendix A).

Qualitative synthesis, which included the qualitative elements of the mixed method papers and was informed by a modified version of the process outlined by Braun and Clark [27], was carried out by the first author and involved repeatedly reading each text to gain a comprehensive understanding of the data. It included identifying notable statements within each text, deriving meanings from these statements, and grouping these meanings to establish key themes. Finally, an extensive description of each theme was developed, following the approach recommended by Aromataris [26]. The qualitative articles were scored as “met” (1), “partially met” (0.5), and “not met” (0), and then they were added to gain a full final score of 10.0–9.00 (high quality), 9.0–7.5 (moderate quality), 7.5–6.0 (low quality), and 6.0 (exclude), as guided by the Critical Appraisal Skills Programme (CASP) checklist (Appendix A).

Further analysis of the data included information regarding participants, sample size, study design, applied measures, type of statistical analysis, evidence levels, and major findings reported. The overall concepts of each paper were analyzed using thematic analysis suggested by [27].

### 3.6. Data Extraction and Analysis

Given the diversity of the data, textual data extraction was undertaken according to best practice principles [28]. Following a modified process outlined by Colaizzi [29], each reviewer (BM, BP, and DT) independently read and re-read the identified articles. Following the independent review, the reviewers exchanged their interpretations of the articles. They identified and aggregated common or recurring patterns among the significant statements and understandings. These patterns were then formulated into thematic representations to describe the phenomena, as proposed by Braun and Clark [27].

## 4. Results

### 4.1. Characteristics of the Included Studies

The literature search yielded 731 potentially relevant publications. Following the removal of duplicates and articles considered to be not on the topic, 229 records remained, and titles were screened for relevance and further review. Of these, 708 were considered directly related to the research question, and full texts were reviewed. A further 212 were removed at this point following the application of inclusion criteria, leaving 17 papers for the final review, as outlined in Figure 1. The 17 studies were undertaken in the USA (n = 9), India (n = 2), China (n = 2), Norway (n = 1), Turkey (n = 1), Korea (n =1), and Sweden (n = 1) (Table 1). 

The findings from the scoping review identified specific factors that assisted students in managing their mental health and well-being during their first-year transition. These are articulated by way of two main themes, which encompass physical practices and training and barriers to success.

### 4.2. Physical Practices and Training

Across the articles in this review, we identified specific physical practices and training that were consistently associated with contemplative practices for both medical and nursing students. Under the umbrella of the theme ‘Physical practices and training’, our focus has been directed toward the fundamental aspects that have surfaced from the comprehensive review process. These pivotal elements have been distilled into two primary subthemes, embodying the crucial factors that are most intimately and persistently linked with contemplative practices suitable for health profession students: elective mind–body courses and mindfulness and yoga practices.

#### 4.2.1. Elective Mind–Body Courses

Within the literature, a significant impact of stress on health professions was identified, especially among undergraduate medical and nursing students. It must be noted the education and training for a medical doctor vary in each country; for example, some programs are direct entry from secondary school, which may be achieved through completing pre-med programs of two general years of university study or even through graduate entry, which is achieved by completing an earlier undergraduate degree [41]. Within this context, depending on the entry pathway or program of study, medical students may experience a different first-year undergraduate student transition than other healthcare students. Needless to say, it is argued these professions require a high level of professional knowledge and skills, leading to higher levels of stress and emotional exhaustion, which were frequently experienced [30,31]. The challenging nature of modern healthcare services signified the need for strategies such as mindful attention awareness techniques, which include yoga, and these were found to positively affect the psychological wellbeing of healthcare students [30,32].

Medical students who participated in an embodied health program for one semester had an initial propensity to be less compassionate towards themselves than other undergraduate students and demonstrated lower empathy scores both before and after the course than previously published data on medical students [31]. The semester-long course consisted of a one-hour-long yoga session followed by a 30 min lecture about the neuroscience of yoga, relaxation, and breathing exercises.

Furthermore, the inclusion of awareness techniques was found to positively impact the psychological well-being of medical students. Decreased stress through mindfulness practices required medical students to be aware of the difficulty of keeping the mind focused, as the distractions of the mind, particularly in the first few weeks of the yoga practice intervention, were challenging to the present-moment experience of awareness. The medical students reported experiencing some space in the last weeks of the program, which included an increased sense of belonging between each other in the course, as medical school was described as an isolating experience for students in an otherwise competitive environment [31].

A positive effect on students’ self-compassion was identified to be statistically significant (*p* = 0.04). In addition, a qualitative examination of students’ reflective notes identified increased self-awareness, as well as new-found self-compassion to accept oneself. Common themes in students’ end-of-course reflections and assessments included a feeling of community in a competitive environment and more empathy for others. These skills were beneficial for health professional students who face heavy academic workloads [31]. High-pressure competitive environments require self-regulation skills, as the stressors can have negative effects on the physical and psychological health of professionals and students [31].

In contrast, an interdisciplinary course offered a semester-long elective course in contemplative neuroscience and yoga attracted a wide range of students, with nearly half from non-science programs, which included business, communication, journalism, arts, and humanities [11]. In the anonymous ratings, students indicated they perceived the course as relevant to their lives and benefitting them personally. The rising popularity of contemplative practices, especially yoga on campuses, is of great interest and relevance to college students [11].

#### 4.2.2. Mindfulness and Yoga Exercises

Within the sub-theme of mindfulness and yoga exercises, seven studies measured mindfulness using the Mindful Attention Awareness Scale (MAAS) [4,11,14,15,33,34]. Mindfulness supports students in their need to find productive ways to develop a clear and realistic understanding of what it means to bring awareness to the present moment itself [14]. In addition, mindfulness and self-compassion were found to be associated with each other, as both were found to improve well-being [30].

The importance of protecting the psychological well-being of healthcare professionals was identified to respond to yoga and contributed to the mental balance of individuals who practiced yoga for a semester [30,32,35]. It was suggested that the nature of modern healthcare services points to the need for mindful attention awareness strategies that include yoga and improve the well-being of healthcare workers [30,31]. Erkin and Aykar [30], in their study, found that yoga integrated into the nursing curriculum increased mindful attention awareness. The study was planned to investigate the effect of a 14-week-long syllabus, including once-a-week 90 min sessions. Students participated in yoga practice, breathing and relaxation exercises, as well as mindfulness meditation with instructional notes and resources [30]. All participants in the study were first-year female nursing students aged between 18 and 21 years. The higher the mindfulness level in students, the higher the level of self-compassion [30]. The feature of self-compassion was identified as one of the important elements of professional identity in healthcare professions [30].

An important element of healthcare professional identity is associated with self-compassion. Self-compassion was described as one of the basic features that nurses should have due to the nature of helping people and maintaining emotional sensitivity to patients [30]. Self-compassion, developed through mindfulness practices, was a powerful predictor of mental health and a recent integration into the undergraduate academic curriculum as a method of maintaining current health and well-being for students [30,36].

In contrast, Kinchen et al. [32] examined stress in nursing students while enrolled in an educational program. The study aimed to deliver an intervention that would lower stress and provide coping skills for nursing students. Students self-selected participation in a one-hour yoga class offered each week for 12 weeks. The results indicate that nursing students experience stress and continue to perceive their lives as stressful regardless of yoga practice [32]. An important finding in the study was that students who practiced the most yoga reported less stress, better quality of life (QOL), greater self-compassion, and higher self-kindness compared to others [32].

Similarly, Kim [35] investigated yogic exercise on life stress and blood glucose levels in nursing students. A significant decrease in stress levels over a 12-week period supports the evidence that yoga exercise can have beneficial effects in reducing psychological problems such as stress in nursing students. In addition, self-compassion was considered one of the important elements of professional identity and central to the provision of quality care. A strong relationship was identified between nurse educators and students, as educators play a central role in helping students manage their stress as they are more in tune with the intensity of academic workload and in a better position to prevent ongoing additional problems in the future [35].

Mind–body intervention research under-pinned program design for non-medical undergraduate students enrolled in either yoga or mindfulness meditation classes for college students [33]. In a study by Gorvine et al. [33], the aim of the research was to compare the effectiveness of yoga or meditation classes in reducing stress for students. The majority of the yoga classes included music, and the meditation intervention topics included were part of the 7-day Mindfulness-Based Stress Reduction (MBSR) professional training materials [42]. The strength of self-compassion on stress was measured in both meditation and yoga interventions. Specifically, it was revealed meditation had a positive impact on self-compassion (F(39) = 16.397), *p* = 0.001), while yoga also had a positive impact, with self-compassion (F(45) = 10.880), *p* = 0.042) [33]. The positive results regarding self-compassion further support including yoga and meditation training in courses designed to improve student well-being.

A study by Qi et al. [34] examined the psychological effects of meditation and breathing-focused yoga practice for non-medical students in China. The aim of the research was to compare energy retention and related stress reduction in a group of undergraduate students. As all yoga classes require a plan of asana sequences to guide the participants, in this study, the structure included breathing and meditation for 10 min (stage 1), 60 min of hatha yoga exercise (stage 2), and 10 min of relaxation (stage 3) which emphasizes the importance of the asana practice in comparison to meditation and relaxation. The attendance rate for all sessions was 100 percent, and the results confirmed that breathing-focused yoga practice is more effective than meditation-focused yoga practice in increasing mindfulness and reducing stress, especially for undergraduate students. Breathing was found to directly increase oxygenation to strengthen the physical body and may easily help undergraduates achieve better attention and awareness (mindfulness).

Tong et al. [14] evaluated the effects of yoga and physical fitness exercises on stress of non-medical undergraduate students. The study aimed to examine the effects of yoga and physical fitness exercises on stress, especially as it is unclear whether yoga or fitness exercise is more effective for stress reduction in this population [14]. Both yoga and fitness exercises helped to increase self-compassion and positive emotions aimed to reduce negative emotions. The results confirmed only yoga exercises had a significant change in both mindfulness improvement and stress reduction [14].

### 4.3. Barriers to Success

The second theme encompassed mental health factors for undergraduate students, which was featured in four articles and associated with the first-year university transition experience [1,3,37,38]. Mental health factors had an impact on undergraduate students during the course of their studies and raised awareness of and responsibilities of college administrators to provide resources that assist students to find balance within their busy academic lives. In this theme, contemplative practices that included modern hatha yoga exercises were identified as providing relief to the physical and mental symptoms of stress experienced by the students.

#### 4.3.1. Emerging Adulthood

In particular, the literature related to stress management for undergraduate students seeks to broaden our understanding of some of the most common stressors among students. These included greater academic demands, new-found independence, financial responsibility, cultivating relationships, and higher-level decision making. University students in emergent adulthood are within a developmental period between late teens to 20s and are distinguished by relative independence from social roles and normative expectations [43]. Managing the mental health and well-being of emergent adults was positively related to individual physical fitness and regular exercise [37].

#### 4.3.2. Stress Management Skills

Mental health, well-being, and quality of life were reported as broad constructs, especially well-being and quality of life, which were often divided into subdomains of physical and psychological well-being or object and subjective measures of quality of life [2]. In order for students to find balance within themselves while at university, yoga interventions for undergraduate populations may be valuable, not only for student mental health but also to improve physical activity outcomes. A study of incoming students from high school participated in an RCT where students were assigned to eight weeks of yoga or Cognitive–Behavioural Stress Management (CMSM) with yoga [3].

The study examined the preliminary feasibility and differential outcomes between CBSM and yoga for first-year female college students. In both cohorts, participants acquired fundamental techniques for stress management, coping mechanisms, cognitive strategies for managing stress, and relaxation skills. Notable differences were observed within the groups in terms of self-regulation (impulsivity and restraint), emotional dysregulation, and interoceptive awareness. Following the eight-week regimen, there was an increase in interoceptive awareness and emotion regulation (*p* = 0.054). Similarly, after eight weeks of CBSM, an increase in restraint was observed (*p* = 0.026). The findings demonstrate that both interventions provide benefits to female students in the first year and a critical time for establishing self-care strategies that protect students during a vulnerable time in their lives [1,3].

Sullivan et al. [37] conducted a study to assess the impact of two distinct yoga styles on mood and salivary cortisol levels in female college students. The study aimed to investigate the susceptibility to stress among millennial women in college. The study was a short-term yoga experience spanning three evening sessions. The intervention involved four study visits: an initial baseline visit, a stretch yoga session, a power yoga session, and a control session. All these sessions took place on consecutive days within a single week. The testing period included a stretch yoga session, a power yoga session, and a control session. The stretch yoga session incorporated elements of meditation, focused breathing, yin poses, final relaxation (savasana), and a closing meditation. Power yoga, on the other hand, emphasizes the intensity of physical postures (asana). Each yoga session was designed to last for 60 min. The findings indicated that both power and stretch yoga significantly enhanced pleasure. The Feel Scale (FS) showed more positive results in the stretch yoga and power yoga conditions compared to the control conditions (*p* = 0.005 and 0.012, respectively). As for the Felt Arousal Scale (FAS), the power yoga condition resulted in higher FAS scores than both the control and stretch yoga conditions (*p* < 0.001). These results suggest that both types of yoga positively influence the psychosocial well-being of female college students [37].

Conversely, Papp et al. [38] conducted a study to examine the impact of a six-week High-Intensity Yoga (HIY) program on various health-related self-reported symptoms such as anxiety, depression, sleep, stress, and subjective health complaints. The HIY exercise regimen included vigorous sun salutations (SS) and physical postures (asanas). The study’s findings revealed no significant differences in any of the health-related outcome measures at the baseline or follow-up after the six-week HIY program. However, the self-reported outcomes indicated that an increase in home training was linked to lower depression scores (*p* = 0.02), improved Pittsburgh Sleep Quality Index (PSQI) scores (*p* = 0.01), and reduced Insomnia Severity Index (ISI) scores (*p* = 0.02). This suggests that home training was associated with fewer depression symptoms, better sleep quality, and fewer insomnia symptoms [38].

#### 4.3.3. The Importance of Practice

The third and final sub-theme identified with the scoping review highlighted that beyond factors that enable and facilitate contemplative practices, barriers to integration were identified and associated with the first-year transition among students. Overall, the key barrier was centered on feeling overloaded. This was a common barrier that had a significant impact on academic functioning. Many students expressed that personal and academic obligations prevented them from attending yoga sessions that, ironically, were designed to reduce stress related to academic expectations [32]. For example, scheduling interventions at the end of the class day was too overwhelming for students who had been on campus most of the day [32]. While university students may greatly benefit from engaging in yoga and/or other forms of physical activity, their school and work schedules may hinder their engagement in these activities [1,32].

## 5. Discussion

This scoping review included 17 articles. The review explored the phenomena of yoga as a mind–body practice that contributed to student well-being and learning in the first-year undergraduate transition to university. Two major themes, ‘physical practices and training’ as well as ‘barriers to success’, were created to explain this phenomenon as it was identified within the literature.

As such, the results of the review focused on strategies that universities might consider to protect and demonstrate their commitment to student well-being. Embodied health courses for medical and nursing students were found to bring meaning and connection [11,31]. Participating as a group within the courses, students experienced a positive effect on connection with each other, as the yoga class required concentration, presence and focus to make the shapes of the asanas. Jacob et al. [44] argued that educational institutions must ensure that students acquire skills such as problem solving, research, interpersonal interactions, and lifelong learning skills. In order to achieve this, a demanding academic curriculum was required which can result in levels of stress among nursing students during the three years of their studies being higher than those among physiotherapy and occupational therapy students [45].

Mindfulness included theoretical definitions such as ‘purposeful attention without making any judgement [30]’ or ‘focusing on the present moment with a sense of openness and nonjudgmental attitude [46]’ and was more directed to the language of training. The most common MBI approaches were based on MBSR principles [46,47]. Various methods of delivery were utilized, including short courses, online delivery, and courses later in the day to assist nursing students in managing stress and fatigue of the day [48].

A combination of yoga, breathing, and mindfulness practices had the effect of developing self-compassion for the participants [30,33]. Self-compassion was found to be an enabling factor in relieving stress, particularly when situated within the mindfulness/awareness tradition. The positive influence of yoga on cultivating self-compassion provides evidence of including more yoga and/or meditation classes to reduce stress in students [32,33]. Similarly, among first-year college students, self-compassion was associated with better well-being, operationalized as subjective vitality [49]. Students with higher scores in self-compassion reported less homesickness, less depression, and high satisfaction with university life [50]. Varela et al. [51] argued the practicing of self-compassion contributed to important insights into one’s own grasping fixation on ego-self, which arise through automatic patterns of conditioned behavior.

Mental Health was identified as a central component of student success in higher education programs. Specifically, the psychological well-being of students was considered central to their success [1,2,52,53,54]. As such, university institutions played key roles in supporting students to cope with their studies [55]. Transition from secondary to higher education was a period of preparation for adjusting to and stabilizing in a new environment, which continues after entry to university and involves change over time [56]. As such, the importance for students to find mental health balance whilst at university was considered important for physical and mental well-being. The results of the review highlight contemplative practices such as yoga, made a difference for students as they searched for ways to regulate the nervous system and balance out some of the associated stress levels with the heightened responsibilities of being a student [57].

Recognizing the levels of stress within student cohorts, yoga studies were effective in developing the construct of internal focus and aimed to interrupt automatic patterns of conditioned behavior, leading to an expansion into other forms of experiencing the body beyond egoic self-focus were beneficial [36]. Shusteman [58] argued that habits are formed over time, embody environmental histories, and can persist even when the original conditions are no longer present, which indicates that will is essentially embodied. Further, Dewey [59] argued it is very hard to see and attend to those things which are closest and familiar to ourselves, requiring greater intelligent self-control and somatic self-awareness. Further, the combination of yoga and breathing was found to be a more effective strategy for reducing stress than yoga and meditation [14,34].

Consistent with the literature, nursing students were found to have higher rates of psychological stress compared to other students, and many experience clinical levels of anxiety and depression [30,39,60]. Sattar et al. [61], for example, in their scoping review, identified positive thinking and active coping, spending time with friends, reflective coping, and physical activity as some of the possible ways to survive the challenging times. A limitation for medical and nursing students was the reluctance to seek help for their mental needs in fear of stigmatization and academic reprisal [62,63,64,65].

Academic overload prevented students from engaging more fully with practices that established resilience and well-being during the first year of higher education. Students identified a lack of time, always being in a rush, a lack of motivation, and excessive required effort [66]. The findings from the identified articles are consistent with the broader literature and support claims that yoga is an exemplary practice for the effective functioning of undergraduate students [4,14,39,40]. It is, therefore, crucial to find those classes of contemplative practices that share an essence and allow the research to make powerful predictions and generalizations about the benefits of yoga for undergraduate students.

### Limitations

Overall, potential limitations of the review may be due to selection bias of the seventeen identified articles in the field of contemplative practices, encompassing yoga and mindfulness practices for undergraduate students. The review was limited to the identified and omitted SCOPUS or the Web of Science (WoS). The search string was formulated in such a way that incorporated yoga with additional terms such as self-compassion, concentration, and attention. It did not include contemplative practices or mindfulness separately. An explicit search for these two additional terms may have altered the results. Nevertheless, our review did identify contemplative and mindfulness practices within the selected Boolean search terms. Further, our review focused only on undergraduate university students, which restricted the cohort and omitted postgraduate students.

## 6. Conclusions

The review highlights the most successful strategies that benefit student well-being were a combination of contemplative practices that laid a foundation for students to experience yoga and mindfulness, primarily as good mental health strategies and suitable for students in the first-year transition. Further research is needed to understand how psychosocial skills developed through contemplative practices afford insight into participatory forms of knowing and emphasize the importance of establishing long-term health goals for young people in their first-year university transition.

### Implications for First-Year University Transition

The scoping review highlighted students’ need to be supported throughout their first-year transition as it is a significant time of transformation and growth. Physical activity improved mental health outcomes. Therefore, educating students about their ability to manage stress through yoga, mindfulness, breathing, and listening to the body contributed to overall well-being. The leading themes within this review resulted in many articles capturing health profession undergraduate students compared to students in other undergraduate disciplines. Higher education policies, along with teaching, learning, and support, must have clearer focus to implement the conditions that encourage and enable students to focus on their well-being during this time of transition in higher education. The onus should be on educators to enable these strategies and on the learner to engage them. Overall, higher education must be realized beyond institutions, including skills and knowledge development for the workplace; institutions should develop, support, and enable good citizens to thrive as they traverse education and as they enter society.

## Figures and Tables

**Figure 1 ijerph-21-00884-f001:**
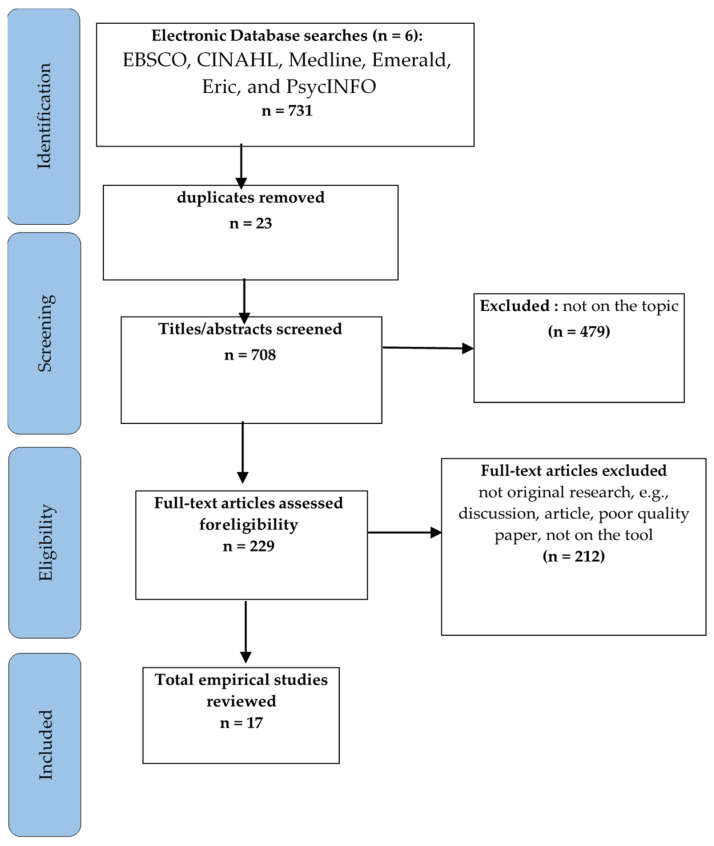
PRISMA flowchart.

**Table 1 ijerph-21-00884-t001:** Included studies.

Study	Design and Aim	Discipline	Sample	Intervention	Findings	Evaluation Tool
Forseth et al. (2022) [1] USA	Quantitative Investigate the feasibility and impact of yoga on student health	University students	n = 12	Hatha or Vinyasa 60 min × 2 8 weeks	Participants had improvements in mental health and increased their levels of physical activity through yoga participation	Perceived Stress Scale (PSS) Beck Depression Inventory (BDI-II)
Park et al. (2017) [3] USA	Quantitative RCT Effects of yoga on cognitive–behavioral awareness	U/G female students	n = 34	Hatha yoga 60 min × 2 8 weeks	Yoga participants improved in interoceptive awareness and emotion dysregulation	Depression Anxiety Stress Scale (DASS-21) Difficulties in Emotion Regulation Scale (DERS); Brief Self-Control Scale Interoceptive Awareness Scale
Elstad et al. (2020) [4] Norway	Quantitative RCT Evaluate yoga’s effect on student distress and other mental health outcomes among healthy university students	U/G students	n = 202	Ashtanga Vinyasa 1.25 h × 2 12 weeks	Yoga reduced psychological distress and sleep problems among students	Hopkins Symptom Checklist (HSCL-25); Bergen Insomnia Scale (BIS); Heart rate variability (HRV); Satisfaction With Life Scale (SWLS); Warwick-Edinburgh Mental Well-Being Scale (WEMWBS); Mindful Attention Awareness Scale (MAAS)
Wolf and Moran (2017) [11] USA	Mixed methods Integrating neuroscience course into health studies	U/G health students	n = 80	Yoga for therapy two-credit points 1 semester	Contemplative practices suggest an interdisciplinary class in health studies is of great interest and relevant	Work intention Mindful Attention and Awareness Scale Depression Anxiety Stress Scale
Tong et al. (2021) [14] China	Quantitative Examine the effects of yoga and physical fitness exercise on stress	U/G students	n = 191	Hatha yoga 60 min 12 weeks	Both types of exercises helped reduce stress	DASS; Mindful Attention Awareness Scale Self-Compassion Scale
Kishida et al. (2019) [15] USA	Quantitative Examine the influence of participating in yoga	U/G students	n = 21	Hatha yoga 150 min 15 weeks	High trait mindfulness subgroup appeared to benefit the most, reporting greater self-compassion	Mindfulness Attention Awareness Scale (MAAS); Self-Compassion-Scale-Short Form (SCS-SF)
Erkin and Aykar (2020) [30] Turkey	Quantitative Investigate the effect of yoga on mindfulness and self-compassion among nursing students	U/G female nursing students	n = 47	Hatha yoga 90 min 14 weeks	A yoga course integrated into the curriculum allowed nursing students to maintain their emotional sensitivity to patients	Mindful Attention Awareness Scale (MAAS) Self-compassion scale (SCS)
Bond et al. (2013) [31] USA	Mixed Methods To experience the effects of a mind–body course	U/G medical students	n = 27	Hatha yoga 60 min 11 weeks	Three themes: 1 increased mindfulness; 2 positive effects on self-regulation; 3 feelings of community	Jefferson Scale of Physical Empathy Cohen’s Perceived Stress Scale Self-Regulation Questionnaire Self-Compassion Scale.
Kinchen et al. (2020) [32] USA	Quantitative Stress in nursing students and self-care interventions into the curricula	U/G nursing students	n = 73	Hatha yoga 60 min 12 weeks	Nursing students experiencing stress continued to perceive their lives as stressful regardless of yoga practice	Perceived Stress Scale-14 (PSS); Self-Compassion Scale (SCS); World Health Organization QOL-BREF (WHOQOL-BREF)
Gorvine et al. (2019) [33] USA	Quantitative Evaluate the effects of yoga and mindfulness meditation on self-compassion, mindfulness, and perceived stress	U/G students	n = 92	Hatha yoga 50 min × 2 10 weeks	The strength of self-compassion was a predictor for stress reduction in both the yoga and meditation interventions	Perceived stress scale 1o (PSS-10) Mindful awareness attention scale (MAAS) Self-compassion scale 9 short form (SCS-SF)
Qi et al. (2020) [34] China	Quantitative RCT Compared the psychological aspect of meditation on breathing-focused yoga on students	U/G students	n = 86	Hatha yoga 80 min 12 weeks	Breathing-focused yoga is more effective than meditation-focused yoga, increasing mindfulness and reducing stress	Work intention Mindful Attention Awareness Scale (MASS) DASS
Kim, S.D. (2014) [35] Korea	Quantitative RCT Yoga plays an effective role in decreasing stress and improving general well-being	U/G nursing students	n = 27	Hatha yoga 60 min 12 weeks	A yoga-based relaxation training program normalizes the functions of the autonomic nervous system	Life Stress Scale for College Students
Cox et al. (2019) [36] USA	Quantitative The mediating roles of body surveillance and body appreciation in relationship to self-compassion	U/G college women	n = 323	Yoga Fit and Power/Vinyasa 75 min × 2 16 weeks	Treating the self with kindness and appreciating the different aspects of one’s body create conditions that cultivate intrinsic motivation, and declines in body surveillance were observed	Neff’s Self-Compassion Scale (SCS) Objectified Body Consciousness Scale Body Appreciation Scale (BAS) Intrinsic Motivation subscale
Sullivan et al. (2017) [37] USA	Quantitative Benefits and effects of power and stretch yoga	U//G female students	n = 33 s	Hatha yoga 60 min × 3	A positive effect on psychosocial well-being and reduced perceived stress	Perceived Stress Scale (PSS); Feeling Scale (FS); Activation–Deactivation Adjective Check List (AD ACL)
Papp et al. (2019) [38] Sweden	Quantitative RCT Investigate yoga for relief from symptoms of anxiety, sleep, depression and stress	U/G students	n = 44	Hatha yoga 60 min 6 weeks	The yoga group had no significant effects on any of the investigated health outcomes	Perceived Stress Scale (PSS) Pittsburgh Sleep Quality Index (PSQI) Insomnia Severity Index (ISI)
Mathad, M.D. (2017) [39] India	Quantitative RCT Evaluate the effectiveness of yoga on psychological health	U//G nursing students	n = 100	Hatha yoga 60 min × 5 8 weeks	Nursing students significantly improved self-compassion and mindfulness in the yoga group	FMI; CD-RIDC 10; Self-compassion scale short form (SCS-SF); SWSL; JSE-HPS; Perceived Stress Scale (PSS)
Bansal et al. (2013) [40] India	Quantitative To improve general and mental well-being	MBBS students	n = 90	Hatha yoga 45 min 5 days per month	Significant improvement in all four areas tested—somatic, anxiety, social dysfunction, and depression	General and mental well-being (GH1-28)

## Data Availability

No new data were created or analyzed in this study. Data sharing is not applicable to this article.

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
