# Peer review of "Yoga as a Contemplative Practice and Its Contribution to Participatory Self-Knowledge and Student Retention: A Scoping Review of the First-Year Undergraduate Student Transition"

_ijerph, 2024, doi:10.3390/ijerph21070884_

Round 1
Reviewer 1 Report
Comments and Suggestions for Authors
The article presents a comprehensive scoping review of contemplative practices that influence participatory self-awareness and student retention during the transition of first-year students in university education. The review is notable for its focus on identifying contemplative practices that can enhance the experience of first-year students and ultimately contribute to their retention in the educational institution.
Among its strengths we can highlight:
Meaningful Focus: The research addresses a critical area in higher education: the successful transition of first-year students. By focusing on contemplative practices, it offers an innovative perspective on how to improve this crucial stage.
Breadth and Depth: The review demonstrates impressive comprehensiveness in the collection and evaluation of relevant literature. The inclusion of multiple studies allows for a comprehensive view of contemplative practices and their impact on student self-awareness and retention.
Practical Implications: The article clearly highlights the practical implications of the findings for educators and university administrators. It provides concrete recommendations for the implementation of contemplative practices in educational settings.
Suggestions for Improvement:
A major shortcoming is detected in the omission of searching the WOS and SCOPUS databases as these are the most commonly used databases for studies published in the field of the manuscript.
Greater Theoretical Contextualisation: It would be useful to provide greater theoretical contextualisation at the beginning of the article to help the reader better understand the conceptual framework on which the research is based.
Consideration of Limitations: A more detailed discussion of possible limitations of the review, such as selection biases or gaps in the literature identified, is suggested.
Overall, the article offers a valuable contribution to the field of higher education by exploring the role of contemplative practices in first-year student transition and retention. With a sound methodology and a clear focus on practical implications, this work has the potential to inform educational policies and practices that promote student success. It is recommended for publication with suggestions for improvement.
Author Response
Reviewer 1
Suggestions for Improvement:
A major shortcoming is detected in the omission of searching the WOS and SCOPUS databases as these are the most commonly used databases for studies published in the field of the manuscript.
Thank you for this question. We worked closely with an academic librarian specialist to ensure accuracy and enable optimal outcomes of the review, which included key database selection. We note the omission of WOS and SCOPUS, however, were guided by the libriaian specialist. Given Google Scholar was also examined for data, we feel this may have identified any additional papers missed by the omission. We have added to the text of regarding this matter for the readers
Page 3, Line 128-133
Greater Theoretical Contextualisation: It would be useful to provide greater theoretical contextualisation at the beginning of the article to help the reader better understand the conceptual framework on which the research is based.
Thank you for this comment, we have added some substantial theoretical discussion for context of the review regarding the conceptual framework in which the research is based.
Page 1, Line 42-50
Consideration of Limitations: A more detailed discussion of possible limitations of the review, such as selection biases or gaps in the literature identified, is suggested.
Thank you for this comment and suggestion, we have added a more explicit limitation section to the paper.
Page 1, Line 42-50
Reviewer 2 Report
Comments and Suggestions for Authors
Thank you for the opportunity to review this paper which is a scoping review of yoga and its impact on college student learning. While prior work has established that mindfulness practices are useful in decreasing college student anxiety and other conditions, this work focuses exclusively on summarizing the evidence base on yoga practice and first year students.
Abstract
The abstract does not mention yoga specifically and led me to believe this was going to be a review of mindfulness practices more broadly rather than yoga.
Background/Lit Review
Succinct and complete.
Purpose.
Rationale for examining yoga only is needed.
Methods
Methods are clear.
Discussion
Complete but for limitations section.
Author Response
Reviewer 2
Abstract
The abstract does not mention yoga specifically and led me to believe this was going to be a review of mindfulness practices more broadly rather than yoga.
Thank you for this comment, however, yoga is mentioned in the abstract three times. We have ensured this is more explicit in the title of the paper for greater clarity.
Page 1, Title
Purpose. Rationale for examining yoga only is needed.
Thank you for this comment and suggestion, we have added a more explicit details regarding the purpose of the study.
Page 1, Line 106-110
Discussion
Complete but for limitations section.
Thank you for this comment and suggestion, we have added a more explicit limitation section to the paper.
Page 1, Line 42-50
Reviewer 3 Report
Comments and Suggestions for Authors The article sent seems interesting to the scientific community. The article generates some important reflections such as the measures taken by university institutions to reduce university dropout. Improving the well-being and self-knowledge of first-year students through yoga as a spiritual, physical and mental discipline is an interesting research objective. In the theoretical framework I would briefly highlight some research that is related to levels of stress, anxiety, and depression in university students during periods of confinement and virtual learning. The qualitative methodology is simple but meets the research objective. The results displayed reflect the search analysis. Table 1: Included studies is interesting because it reflects in detail the characteristics of the research examined. The authors have to expand more broadly the section: “Implications for first-year university transition”
Author Response
Reviewer 3
In the theoretical framework I would briefly highlight some research that is related to levels of stress, anxiety, and depression in university students during periods of confinement and virtual learning.
Thank you for this suggestion, we have added additional background regarding the level of stress, anxiety, and depression among university studies. This helps to support and further clarify the background to the review.
Page 3, Line 102-104
The qualitative methodology is simple but meets the research objective.
Thank you, no changes made
The results displayed reflect the search analysis. Table 1: Included studies is interesting because it reflects in detail the characteristics of the research examined.
The authors have to expand more broadly the section: “Implications for first-year university transition”
Thank you for this comment and suggestion, we have added a more explicit implications within the paper to address this comment.
Page 16, Line 561-573
Reviewer 4 Report
Comments and Suggestions for Authors
Dear authors,
These are my comments referring to your work:
Introduction: This section presents, coherently, the role of contemplative pedagogy as yoga and mindfulness practices for university students and . Then, in this section are presented some studies that highlighted the previous results of contemplative practices for university students. Moreover, the introduction is aligned with the objectives. Although, I consider that it isn't necessary to create subchapter 1.1. Benefit of contemplative practices for university students and health trainees.
Purpose of the study and Methods: are well explained. One note: explain what means "BJI".
Results: I suggest to re-name the subchapters because the present division is confusing. It is necessary a better organization, so that on the one hand, name the type of the contemplative education programs and, on the other hand, name the principal effects of the contemplative pedagogy following other criteria (it isn't suitable to divide the effects into: "health professions" and "mental health").
Discussion: I don't understand the relevance of the lines 443-448 in this section.
Conclusion: It requires an improving with the specification of the benefits in mental health domain regarding the contemplative programs at the university students.
Kind regards.
Author Response
Reviewer 4
Introduction:
Although, I consider that it isn't necessary to create subchapter 1.1.
Thank you for this comment and suggestion, we have deleted this sub-heading
Purpose of the study and Methods
One note: explain what means "BJI".
Thank you for this comment, we have added the full name of the Joanna Briggs Institute and its acronym within the text (JBI)
Page 4, Line 173
Results:
I suggest to re-name the subchapters because the present division is confusing. It is necessary a better organization, so that on the one hand, name the type of the contemplative education programs and, on the other hand, name the principal effects of the contemplative pedagogy following other criteria (it isn't suitable to divide the effects into: "health professions" and "mental health").
Thank you for this comment and suggestion, we have changed the headings to reflect the feedback and make this clearer for the readership and to address the comment.
Pages 6 and 9, Line 241-243 and 371
Discussion:
I don't understand the relevance of the lines 443-448 in this section.
Thank you for this comment, this section is present to emphasize how nursing students experience a high degree of stress as they are required to participate in course placements at a hospital as well as the usual course university examinations. We have sought to make this more explicit within the text
Page 11 Line 449-454
Conclusion:
It requires an improving with the specification of the benefits in mental health domain regarding the contemplative programs at the university students.
Thank you for this suggestion, we have added to the conclusion to reflect the feedback and make this more explicit.
Pages 16, Line 551-557
Reviewer 5 Report
Comments and Suggestions for Authors
Provided a generally sound overview regarding the scooping review and the processes involved. It would be important to note an important difference between undergraduate Nursing education and Undergraduate Medical education. While Medication education programs are considered undergraduate ...it is important to note in the article that most Med. schools require at least 2 years of undergraduate courses before being eligible for the Medical Education program. Therefore, Medical students experience a different first-year undergraduate student transition than Nursing students which should be noted in the article.
The rest of the findings from the scoping review were generally reasonable.
Comments on the Quality of English LanguageThe English language was generally reasonable; however; the grammar, punctuation, and abbreviations need to be checked throughout the paper.
Author Response
Reviewer 5
Medical education. While Medication education programs are considered undergraduate ...it is important to note in the article that most Med. schools require at least 2 years of undergraduate courses before being eligible for the Medical Education program. Therefore, Medical students experience a different first-year undergraduate student transition than Nursing students which should be noted in the article.
Thank you for this comment. We agree and have added to this section of the paper to more explicit.
Page 6 Line 234-238
Pages 7, Line 254-260
The English language was generally reasonable; however; the grammar, punctuation, and abbreviations need to be checked throughout the paper.
Thank you for this comment. Although we are all native speakers of English and it is our only mode of communication verbally or in writing, we have had the paper reviewed from outside the authorship team to ensure grammar, syntax and punctuation is correct and presentable.
See throughout the document